# Effects of NaCl on the Physical Properties of Cornstarch–Methyl Cellulose Blend and on Its Gel Prepared with Rice Flour in a Model System

**DOI:** 10.3390/foods12244390

**Published:** 2023-12-06

**Authors:** Juhee Kim, Yoon Hyuk Chang, Youngseung Lee

**Affiliations:** 1Department of Food Science and Nutrition, Dankook University, Cheonan 31116, Republic of Korea; rlawngml6078@naver.com; 2Department of Food and Nutrition, and Bionanocomposite Research Center, Kyung Hee University, Seoul 02447, Republic of Korea

**Keywords:** NaCl, cornstarch–methyl cellulose (CS–MC) mixtures, physical properties, rice flour, 3D food printer

## Abstract

This study investigated the impact of NaCl on the physical properties of cornstarch–methyl cellulose (CS–MC) mixtures and their gels prepared with rice flour in a model system. Opposite trends were observed, showing that NaCl led to decreased viscosity of the CS–MC mixtures (liquid-based), whereas a more stable and robust structure was observed for the rice-flour-added gels (solid-based) with the addition of NaCl. The interference of NaCl with the CS-MS blend’s ability to form a stable gel network resulted in a thinner consistency, as the molecules of the CS-MS blend may not bind together as effectively. On the contrary, NaCl showed the potential to enhance the protein network within CS–MC gels prepared with rice flour, thereby contributing to an augmentation in the stability or firmness of the cooked gels. Careful utilization of NaCl to optimize the physical properties of the CS–MC blends, as well as the gels based on rice flour, should be performed.

## 1. Introduction

Starch, a carbohydrate substance primarily stored in plants, represents the most significant food resource, accounting for 70–80% of the world’s calories consumed by humans [1]. In addition to its nutritional value, starch exhibits various physical properties such as swelling, gelatinization, gelation, and retrogradation. It has been widely utilized as a thickening agent, gelling agent, and stabilizer in the food industry [2,3]. Corn is one of the three most important food grains in the world, alongside wheat and rice [4]. The demand for corn and its derivatives, such as flour and starch, is increasing because it is recognized as a valuable food grain [5,6]. Cornstarch (CS), extracted from the endosperm of corn (*Zea mays* L.), is widely utilized worldwide as a primary source of starch due to its cost-effectiveness and ease of extraction in contrast to rice starch [7,8].

However, native starch is known to have limited applications in food processing due to certain defects, including sensitivity to shear, heat, and acid associated with the processing conditions. Namely, native starch can be sensitive to shear forces. Excessive mechanical processing or shear can lead to breakdown and a reduction in viscosity. Additionally, it may not withstand high temperatures without undergoing irreversible changes such as gelatinization or retrogradation [9,10].

To enhance these undesirable properties, chemical and enzymatic modification methods have been employed. However, due to their high cost and processing disadvantages, researchers have recently focused extensively on exploring easier, eco-friendly, and more cost-effective physical methods [11].

Hydrocolloids are polysaccharides derived from plants (i.e., biomolecules) and are among the most commonly used substances in food preparation [12,13]. They bind with water to increase the viscosity of the mixture and have a high molecular weight, making them commonly used as thickeners and gelling agents [14,15]. Among them, methyl cellulose (MC) is the least modified form, with a methoxy group instead of a hydroxyl group [16]. It has found extensive use in the food industry due to its effective ability to modify and control the physical properties of starch [17]. Both starch and hydrocolloids influence the rheological properties of food, such as viscosity enhancement, water binding, texture modification, gel formation etc., and their impact is significantly influenced by factors such as type and concentration, the ratio of amylose to amylopectin, pH, and the presence of other components like sugars or salts [18].

The addition of salt can have several effects on the gelatinization and rheological properties of native starches and their pastes, depending on factors such as the type of starch, concentration of salt, and the overall composition of the food system. Hence, it is important to understand the gelatinization and rheological characteristics of the mixtures of starch and hydrocolloid in the presence of salts. This understanding is essential for the development of processes and products, encompassing aspects like process design, quality, and shelf-life extension [19].

A number of studies on the rheological and gelatinization properties of various starch–hydrocolloid mixtures in the presence of salts have been conducted, showing that the addition of salts influenced the rheological and gelatinization characteristics of mixtures containing starch, hydrocolloid, and salt. These effects also varied depending on the type and concentration of the salt. Some examples include CS–carrageenan gum, CS–guar gum, rice starch–xanthan gum (XG), CS–XG mixtures, and pea starch–XG [20]. However, there is limited information available regarding the gelatinization and rheological properties of a mixture consisting of CS and MC in the presence of NaCl.

Rice (*Oryza sativa* L.) holds significant importance as a staple food for populations in numerous countries, particularly in Asian countries. Presently, the Korean food industry is venturing into the use of rice in new processed food products intended for both domestic consumption and export [21]. Rice is mostly ground into rice flour (RF) for processing purposes [21]. Furthermore, RF has been recognized as an excellent alternative to wheat flour, particularly for those with gluten intolerance. Consequently, diverse rice flour-based products have been developed.

Therefore, as the effect of salt varies depending on the type of starch and hydrocolloid, it is warranted to study the properties of the CS–MC mixture and its interactions with RF in food model systems [22,23]. To the best of the authors’ knowledge, no comprehensive study has been conducted on the effect of salt on the physical properties of CS–MC mixtures and the gels prepared with RF in a model system.

3D (three-dimensional) food printing is an additive manufacturing technology for food production that utilizes computer-aided design (CAD) software (Cura 2.4, Ultimaker BV, Geldermalsen, The Netherlands) without the need for additional tools. This technology has the capacity to enhance the functional and structural properties of food by utilizing various materials and creating diverse shapes [24,25]. It is particularly important to discover optimal conditions for improving printability, as the applicability of 3D food printing expands when it becomes possible to manufacture complex structures [26]. In addition, 3D food printing can be effectively used to assess the dimensional stability of semi-solid or solid products such as gels [27].

In this study, we investigated the impacts of NaCl on the physical properties of CS–MC mixtures in Experiment 1 and on its gel prepared with RF in a model system in Experiment 2. We believe that the first experiment represents a fundamental study of the physical properties of CS–MC blends. In contrast, the second experiment is more aligned with the practical application based on these blends. This is evident as solid gels prepared with RF and water, incorporating the same CS–MC blends, were produced using 3D printing.

## 2. Materials and Methods

### 2.1. Experiment 1

#### 2.1.1. Materials

The ingredients used to prepare the mixtures were cornstarch (Fresco Co., Ltd., Gimpo, Republic of Korea), Methyl cellulose (MC) (Jupiter International Co., Ltd., Seoul, Republic of Korea) and NaCl (ES Food Co., Ltd., Gunpo, Republic of Korea). In addition, for 3D printing, rice flour (RF) with 100% rice content (Sungjin Food Co., Ltd., Gwangju, Republic of Korea) was sourced. Deionized water (Milli-Q Direct 16 System, Merck, Darmstadt, Germany) was utilized in this study.

#### 2.1.2. Preparation of CS–MC Mixtures

The CS–MC dispersions were prepared at a concentration of 5% (*w*/*w*) with a mixing ratio of 17:0.5 (CS/MC) in the presence of 1.0%, 2.0%, 3.0%, and 4.0% (*w*/*w*) NaCl levels. The ratio of CS to MC was determined empirically for 3D printing application based on a preliminary experiment, following the methodology outlined by Ko et al. [27]. The specific ratio of CS and MC was determined based on preliminary experiments, showing the most stable rheological properties for 3D printing within the practical range of use for the CS–MC blends. A control mixture without salt was also prepared. Each dispersion was thoroughly stirred for 1 h at 25 °C using a magnetic stirrer. Subsequently, the mixtures were heated for 30 min at 95 °C in a water bath (PS-1000, Eyela Co., Ltd., Shanghai, China) with gentle agitation provided by a magnetic stirrer to prevent sedimentation and lump formation. After heating, the hot pastes were immediately transferred to a rheometer (MCR-012, Anton Paar Co., Ltd., Graz, Austria) equipped with a plate-to-plate system (diameter: 5 cm, gap: 1 mm) for the measurement of steady and dynamic shear rheological properties.

#### 2.1.3. Swelling Power

The swelling power was assessed by modifying the method outlined by Yoo et al. [28]. The swelling power indicates the capacity of starch granules to take in water and the degree to which amylose is released during the swelling process. Dispersions of CS–MC at a concentration of 5% (*w*/*w*), containing 1.0%, 2.0%, 3.0%, and 4.0% (*w*/*w*) NaCl levels, were prepared. The CS–MC–NaCl dispersions were stirred for 1 h at room temperature and then heated for 30 min at 95 °C in a water bath, as previously described. Subsequently, the hot paste was rapidly cooled to room temperature in ice water for 5 min. To determine the swelling power, the pastes were centrifuged at 3000 rpm for 20 min using a centrifuge (Avanti J-E, Beckman Coulter Inc., Seoul, Republic of Korea). The supernatant was carefully removed, and the weight of the sediment was measured. The ratio between the weight of the sediment and the weight of the dry sample (g/g) was calculated as the swelling power, following the method described by Wang et al. [29]. This calculation assumed that the total amount of methyl cellulose remained in the supernatant. All measurements were performed in triplicate.

#### 2.1.4. Pasting Properties

The pasting properties of CS–MC–NaCl mixtures were assessed using a Rapid Visco Analyzer (RVA-4, Newport Scientific Inc., Warriewood, Australia). Dispersions of CS–MC at a 5% (*w*/*w*) concentration, containing 1.0%, 2.0%, 3.0%, and 4.0% (*w*/*w*) NaCl levels, were prepared as samples. Subsequently, 28 g of each dispersion was poured into an aluminum canister, stirred manually for 30 s using a plastic paddle, and then placed into the RVA. The pasting profile of the sample was monitored following a standard program (STD1): Initially held at 50 °C for 1 min, heated from 50 to 95 °C at a rate of 12 °C/min, and then maintained at 95 °C for 2.5 min. Subsequently, the hot sample was cooled to 50 °C at a rate of 12 °C/min and held at 50 °C for 2 min. The rotation speed of the paddle started at 960 rpm, ensuring uniform dispersion for the first 10 s, and was then maintained at 160 rpm for the remaining duration. Using the RVA curve, various pasting properties, including peak viscosity, trough viscosity, breakdown viscosity, final viscosity, setback viscosity, and pasting temperature of the CS–MC–NaCl mixtures, were determined. All measurements were performed in triplicate.

#### 2.1.5. Rheological Measurements

The rheological properties of the CS–MC–NaCl paste were assessed using a rheometer, with measurement conditions established following the method outlined by Gil and Yoo [20]. The paste was placed onto a rheometer plate at 25 °C using a measuring spoon with a volume of 1.25 mL. Steady shear flow data were obtained over a range of shear rates from 0.1 to 1000 s^−1^. To characterize the steady shear rheological properties, the experimental data were fitted to the Power law and Casson models as follows:

Power law model: σ = K·γ^n^
(1)


Casson model: σ^0.5^ = K_oc_ + K_c_γ^0.5^
(2)

where σ is the shear stress (Pa), γ is the shear rate (s^−1^), n is the flow behavior index, K is the consistency index (Pa·s^n^), and the square of the intercept (K_oc_)^2^ is the Casson yield stress (K_oc_). The apparent viscosity (η_a,500_) at a shear rate of 500 s^−1^ was calculated using the magnitudes of K and n.

Dynamic viscoelasticity was obtained from the frequency sweeps over the range of 0.63–62.8 rad/s at the 1% strain, which was in the linear viscoelastic region. To determine the linear viscoelastic region, a deformation sweep test was performed with a strain range of 0.01 to 100% at a constant frequency of 6.3 rad/s. The dynamic shear rheological properties, which contained the storage modulus (G′), loss modulus (G″), complex viscosity (η*), and tan δ (G″/G′) were determined. Further, all rheological measurements were performed in triplicate. All measurements were performed in triplicate.

### 2.2. Experiment 2

#### 2.2.1. Preparation of CS–MC Pastes

The terms paste and gel are defined as follows: The term ‘paste’ refers to the samples printed using a 3D printer and not cooked, while the term ‘gel’ refers to the samples after the printed pastes were cooked. A CS–MC paste was prepared with slight modifications following preliminary trials in accordance with the method outlined by Ko et al. [27].

The paste was prepared to contain various concentrations (1.0%, 2.0%, 3.0%, 4.0%) of NaCl, and the content of each ingredient (per 100 g) is presented in Table 1. Control mixtures without salt were also prepared. Initially, CS and NaCl were dissolved in distilled water and manually mixed for 5 min at room temperature (approximately 25 °C). Subsequently, RF and MC were thoroughly blended into the starch solution until a uniform paste was achieved. The CS–MC paste was then stored in a refrigerator at 4 °C for 24 h to allow for adequate moisture absorption (i.e., rehydration).

We wanted to produce a simple solid-based gel, like a model system, for the practical application using the MC-CS blends. Among the many candidates to help produce a solid-based sample, we thought RF could be a good candidate as it is a versatile ingredient commonly used in the processing of various food products.

#### 2.2.2. 3D Printing Process

Two different shapes of starch paste printed using a 3D printer were employed in this study. The cuboid-shaped paste was utilized for assessing printing accuracy and underwent post-cooking, including all textural analyses. Conversely, the cylindrical-shaped paste was exclusively employed for dimensional stability measurements. The 3D printing of starch paste with various concentrations of NaCl was conducted using a 3D food printer (YOLI-LAB, YOLILO Co., Ltd., Seoul, Republic of Korea), following the methods described by Ko et al. [24] with slight modifications, particularly in the printing parameters such as nozzle dimension, infill level, and pattern. The paste was loaded into a 50 mL syringe, and the 3D design file (.stl format) used a cuboid shape with dimensions of length/width: 30 mm and height: 10 mm. The nozzle diameter was 1.1 mm, and the printing parameters were set as follows: a nozzle height between layers of 1.8 mm, a first layer height of 1.7 mm, an infill level of 80% using a rectilinear infill pattern, and a nozzle speed of 30 mm/s.

It is generally known that 3D printing the nozzle or layer height at 80% or equal to the nozzle diameter allows for good deposition and adhesion of the deposited filaments. However, we conducted numerous preliminary tests to achieve stable starch gels, leading us to the conclusion of using a specific ratio between the nozzle height (1.8 mm) and the nozzle diameter (1.1 mm). This decision was based on the close proximity between the starch paste printed by the 3D printer and its nozzle. The proximity was such that the nozzle could touch the printed paste, causing it to collapse from its original shape. In this study, successful printing was defined as the ability to maintain the original shape without collapsing for up to 20 min after printing.

#### 2.2.3. Dimensional Stability

Dimensional stability was assessed following the methods outlined by Kim et al. [30], with slight modifications. A cylindrical shape with a diameter of 25 mm and a height of 40 mm, utilizing a 100% infill level with a concentric infill pattern, was employed for the experiment. The dimensional stability was monitored by comparing the height of the deformed shape after printing with the height of the 3D template, and it was calculated as follows:$$\mathrm{Dimensional}\;\mathrm{stability}\;(\%)=\frac{\left(\mathrm{Target}\;\mathrm{value}-\mathrm{Measured}\;\mathrm{value}\right)}{\mathrm{Tatget}\;\mathrm{value}}\times 100$$

All measurements were performed in triplicate.

#### 2.2.4. Printing Accuracy

Printing accuracy was determined by comparing the length, width, and height values of the final printed product with those obtained from a preset 3D template, following the methods outlined by Fan et al. [31]. The paste was 3D-printed into a cuboid shape with a length of 25 mm and a width of 15 mm, using a 50% infill level and an aligned-rectilinear infill pattern. The printed product was stored at 25 °C, and the printing accuracy was calculated as follows:$$Printing\;accuracy\;(\%)=\frac{Measured\;value}{Tatget\;value}\times 100$$

All measurements were performed in triplicate.

#### 2.2.5. Post-Processing Characteristics

The 3D-printed pastes with a cuboid shape were baked at 120 °C for 20 min in an electric oven (EON-B303M, SK Magic Co., Seoul, Republic of Korea). After cooking, the samples were allowed to cool for 10 min at room temperature. All measurements were conducted at 25 °C.

##### Cooking Loss

Cooking loss was assessed following the methods outlined in previous work [30]. After the 3D-printed sample underwent post-processing as described above, any surface water on the sample was gently wiped off, and then the sample was weighed. Cooking loss was calculated by comparing the weight of the sample before (*C*0) and after cooking (*C*1), as follows:$$Cooking\;loss\;(\%)=\frac{C0-C1}{C0}\times 100$$

All measurements were performed in triplicate.

##### Shrinkage

Shrinkage ratios were determined following the methods outlined in previous research [30]. The transversal and longitudinal shrinkage of the sample were measured as a percentage reduction in the parallel and vertical lengths of the cooked sample, respectively, in comparison to the uncooked sample. The maximum length of the uncooked and cooked samples was employed in the calculation. The equation used for the calculation is as follows:$$\mathrm{Transversal}\;\mathrm{shrinkage}\;\mathrm{ratio}\;(\%)=\frac{T0-T1}{T0}\times 100$$
$$\mathrm{Longitudinal}\;\mathrm{shrinkage}\;\mathrm{ratio}\;(\%)=\frac{L0-L1}{L0}\times 100$$
where *T*0 and *T*1 represent the parallel diameters of the uncooked and cooked samples, and *L*0 and *L*1 represent the vertical diameters of the uncooked and cooked samples, respectively. All measurements were performed in triplicate.

#### 2.2.6. Mechanical Properties

##### Texture Profile Analysis

The sample, in a cuboid shape with a length of 30 mm and a width of 10 mm, was 3D-printed and used after post-processing. Texture profile analysis (TPA) was conducted using a texture analyzer (TA-XT2, Stable Micro Systems Co., Haslemere, UK) equipped with a P/50 cylindrical probe (diameter: 50 mm). The measurement conditions were determined with reference to previous studies by Chen et al. [32] with slight modifications: a pre-test speed of 3 mm/s, a test speed of 1 mm/s, a post-test speed of 3 mm/s, compression to 50% strain, and a trigger force of 5 g. The samples underwent compression for two consecutive cycles, and parameters such as hardness, adhesiveness, cohesiveness, springiness, and chewiness were determined from the TPA graphs. These parameters were used to characterize the textural properties of the 3D-printed samples. Each sample was measured in five replicates.

##### Cutting Test

A cutting test was conducted following the method described by Zhou et al. [33] with slight modifications. Similar to the TPA test, a cuboid-shaped sample (length and width: 30 mm, respectively; height: 10 mm) was used in the experiment. A shear force was calculated using a texture analyzer equipped with an A/LKB blade probe. The sample was measured under the following conditions: a compression distance of 10 mm and a test speed of 1 mm/s. The shear force was expressed as the maximum force (in grams) required to cut the sample, as observed on the force-time curve. This measurement is commonly used to evaluate the hardness of foods. All samples were measured in five replicates.

#### 2.2.7. Statistical Analysis

All experimental results are presented as mean ± standard deviation (SD). The data were analyzed using analysis of variance (ANOVA) in XLSTAT software version 2012 for Windows (Adinsoft Inc., Paris, France), and significant differences between samples were assessed using Tukey’s test (*p* < 0.05).

## 3. Results and Discussion

### 3.1. Experiment 1

#### 3.1.1. Swelling Power

The swelling power of the CS–MC mixture with the addition of various NaCl concentrations (0%, 1.0%, 2.0%, 3.0%, and 4.0%) ranged from 5.5 to 24.8 (g/g), as shown in Table 2. It was observed that as the NaCl concentration increased, the swelling power of the CS–MC mixture significantly decreased. Gil and Yoo [20] also reported that the addition of NaCl led to a significant reduction in the swelling power of sweet potato starch (SPS)–xanthan gum (XG) mixtures. This observation was in line with the findings of Samutsri and Suphantharika [34], showing that the addition of NaCl significantly decreased the swelling power of rice starch–xanthan gum mixtures. This decrease in swelling power can be attributed to the inhibitory effect of salt on starch swelling and can be explained by the following mechanisms: (1) interference of ions based on salt type and (2) competition for available water between starch and salt [23].

The polarity and charge strength of the salt ions have a significant impact on the breakdown of hydrogen bonds between starch molecules, allowing starch to absorb water and swell. However, ions with relatively small or symmetrical structures tend to exhibit low polarization, providing protection to starch molecules and inhibiting moisture absorption, thereby reducing swelling power [29]. In the case of NaCl, it ionizes into Na^+^ and Cl^−^ ions in water, which interfere with the interaction between water and starch molecules, inhibiting swelling [35,36]. Additionally, salt competes with starch for water molecules and exhibits a strong water-holding capability (WHC), attracting more water molecules [37]. Consequently, the availability of water to starch in the presence of NaCl is reduced, leading to the suppression of starch granule swelling. This competition with starch for moisture absorption also contributes to the inhibition of granule swelling.

#### 3.1.2. Pasting Properties

The pasting parameters of the CS–MC mixture with different concentrations of added NaCl are presented in Table 3. The addition of NaCl led to a gradual and significant decrease in peak, trough, and breakdown viscosity, with variations depending on the concentration. In order to increase viscosity, starch granules typically need to absorb a sufficient amount of moisture and swell [38]. However, as mentioned earlier, salt inhibits starch swelling, making it challenging for the starch mixture containing salt to achieve high viscosity. Furthermore, chloride ions (Cl^−^) from NaCl can bind to the positively charged regions of starch molecules on the starch granules, delaying granule swelling and reducing suspension viscosity [39].

The pasting temperature was significantly higher than that of the control (0% NaCl), which is also related to the inhibitory effect of salt on swelling. This observation was more pronounced in the samples with 2% NaCl or higher.

Similar observations have been reported for a sweet potato starch–xanthan gum mixture [20]. Breakdown viscosity, indicating the disruption of swollen starch granules under continuous heat and shear, is an important quality evaluation parameter during processing. With increasing salt concentration, the breakdown viscosity decreased, suggesting slower destruction of swollen starch granules and greater heat resistance. Therefore, adding salt to the starch–hydrocolloid mixture can stabilize the starch structure and is preferable when the processing involves heat application. Setback viscosity, which reflects the degree of retrogradation, showed no significant difference among the samples.

#### 3.1.3. Rheological Properties

##### Steady Shear Flow Properties

All mixtures exhibited a gradual decrease in the rate of shear stress increase as the shear rate increased, indicating shear-thinning behavior. Moreover, the addition of NaCl led to an increase in shear stress (Figure 1), but there were no significant differences in viscosity among the samples.

Table 4 presents the rheological parameters obtained by fitting the experimental data to the power law model and the Casson model. The flow behavior index (n) values (ranging from 0.22 to 0.34) for the CS–MC–NaCl mixture were slightly higher than the n-value (0.20) for the control. In all samples, the n-value was less than 1, indicating that the CS–MC–NaCl mixture exhibited pseudoplastic behavior, behaving as a fluid with shear-thinning properties. Notably, significant differences in n-values were observed among samples depending on the amount of added NaCl. In the CS–MC–NaCl complex system, increasing the NaCl concentration resulted in higher n-values, indicating reduced plasticity of the mixture. In other words, the addition of NaCl decreased the shear-thinning behavior of the CS–MC mixture compared to the control. Shear-thinning properties arise as a result of particle alignment in the direction of constant flow under the applied shear stress or particle breakdown [20]. This observation was in good agreement with Gill and Yoo [20], who reported that the n-values of the SPS–XG–NaCl mixtures were slightly higher than those of the control (0% NaCl). They indicated that the shear-thinning of SPS–XG–NaCl mixtures in the presence of NaCl was decreased compared to the control.

The consistency index (K), along with the flow behavior index (n), is used to characterize the flow behavior of non-Newtonian fluids. The consistency index and yield stress for the CS–MC mixtures, determined using the power law and Casson models, were found to be lower than those of the control. Additionally, these values exhibited a decline as the NaCl concentration increased (Table 4), implying that the addition of NaCl led to reduced flow resistance, as reported by Gil and Yoo [20]. This observation was in line with Gill and Yoo, who noted that the consistency index and yield stress of the SPS–XG–NaCl mixtures were much lower than that of the control. In addition, these values decreased with an increase in the NaCl concentration. They attributed this observation to the addition of NaCl, which allowed resistance to flow.

Choi and Yoo [40] reported the synergistic effect of XG on starch viscosity. However, this effect may be less pronounced when adding NaCl to the mixture due to the conformational change in the XG in the presence of NaCl [41]. We believe that a similar mechanism for the MC in the presence of NaCl resulted in a less effective synergistic effect on CS viscosity.

##### Dynamic Viscoelastic Properties

Figure 2 illustrates the dynamic viscoelastic properties of the CS–MC mixtures at varying frequencies. All CS–MC–NaCl mixtures exhibited a higher storage modulus (G′) than the loss modulus (G″) within the tested frequency range (0.63–62.8 rad/s), indicating a gel-like behavior [42]. Additionally, both G′ and G″ increased with increasing frequency. Gill and Yoo [20] also reported that SPS–XG mixtures, with the addition of NaCl, exhibited an increase in the magnitudes of G′ and G″ with an increase in ω, and G′ was notably higher than G″.

Table 5 presents the storage modulus (G′) and loss modulus (G″) at 6.28 rad/s for the CS–MC mixtures with different NaCl concentrations. The addition of salt resulted in a higher dynamic viscoelastic modulus compared to the control, and this increase was more pronounced with higher NaCl concentrations (3.0% and 4.0%) than with lower concentrations (1.0% and 2.0%). Furthermore, NaCl addition had a greater impact on G′ than on G″ based on their values. This suggests that NaCl primarily influences the elastic properties rather than viscosity, thereby stabilizing the CS–MC mixture. In general, the addition of salt inhibits electrostatic repulsion, leading to improved intermolecular interactions between starch and hydrocolloids. This interaction reduces the hydrodynamic size of molecules and promotes network formation [43]. The resulting network formation enhances the elasticity of the mixture, which is supported by the higher G′ compared to G″ observed in this study.

Typically, the dynamic moduli (G′ and G″) of MC in the presence of NaCl may experience an elevation caused by the self-aggregation of hydrocolloid molecules, as observed by Meyer et al. [44]. The addition of salts can strengthen the intermolecular interaction among hydrocolloid molecules because of the charge-screening effect [43]. This effect leads to a decrease in the hydrodynamic size of the molecule, ultimately facilitating the formation of a network. Such network formation is associated with an augmentation in the elastic properties of CS–MC–NaCl mixtures, as evidenced by the higher G′ in comparison to G″.

Complex viscosity is a measure of resistance to deformation resulting from a combination of viscosity and elasticity [45]. As shown in Table 5, as the concentration of NaCl increased, the complex viscosity also significantly increased, indicating greater resistance. This suggests that NaCl imparts stability against external forces. The tan δ value reflects the viscoelastic behavior of semi-solid foods. When tan δ is less than 1, it indicates that the sample is predominantly elastic [46]. In this study, all samples had tan δ values ranging from 0.17 to 0.20, and these values tended to increase with frequency. Based on these findings, it is believed that the reduction in tan δ values for the CS–MC mixtures in the presence of NaCl is likely attributed to the self-aggregation of MC molecules induced by salt. This phenomenon seems to have a more significant impact on the elastic properties of CS pastes.

### 3.2. Experiment 2

#### 3.2.1. Dimensional Stability

Figure 3 illustrates the dimensional stability of the CS–MC–NaCl pastes. To effectively assess the impact of NaCl concentration on printing performance and dimensional stability, the height of the 3D-printed sample was set at 40 mm. The deformation rate of the CS–MC mixtures significantly decreased as the NaCl concentration increased (Figure 3). In contrast, the control group exhibited low stability, with a deformation rate of 7.08%. This was due to inadequate vertical stacking during printing, resulting in a gradual collapse of the shape. Moreover, weak interlayer bonding in the control led to the formation of empty spaces on the surface, preventing the accumulation of layers and resulting in uneven lines.

As shown in Figure 4, the CS–MC pastes with NaCl additions maintained their shape throughout the printing process, resulting in a smooth surface with well-defined lines (i.e., high resolution). The dimensional stability of the 3D-printed pastes varied depending on the NaCl concentration. Notably, the CS–MC paste with a 4% NaCl concentration exhibited optimal rheological properties with a low deformation rate (0.42%), indicating high dimensional stability and successful printing.

#### 3.2.2. Printing Accuracy

Printability can be significantly influenced by material properties, including fluidity for smooth extrusion through the nozzle and the ability to retain shape after printing [25]. Notably, a decrease in the extrusion properties of the paste can lead to challenges in uniformly extruding layers, resulting in a critical loss of resolution in 3D-printed products [47]. In this experiment, a cube filled to 50% was chosen as the test shape because it allowed for the observation of shape collapse during printing and the formation of filling lines.

Figure 5 illustrates a 3D-printed product in the aforementioned shape using the CS–MC paste with varying concentrations of NaCl. It was observed that the addition of NaCl improved the accuracy of 3D-printed samples, showing that the filling lines within the printed cubes appeared most stable and clean-cut in the case of the 4% NaCl sample. It is believed that the presence of NaCl interfered with starch moisture absorption, rendering the paste softer and facilitating smooth extrusion through the nozzle. This resulted in high printing accuracy without loss of shape [48,49].

The printing accuracy of the CS–MC–NaCl gel is presented in Table 6. The control group experienced shape distortion and collapse, with increased length and width and decreased height. In contrast, the samples with NaCl additions, except for a slight distortion at a low concentration (1% NaCl), showed no significant spread or collapse. This suggests that the addition of NaCl can significantly enhance the printing accuracy of the paste. It was also observed that the standard deviation for the length, width, and height values of the final printed product tended to decrease as the concentration of NaCl increased. This suggests that the addition of NaCl contributed to increased stability in the printed products. Moreover, the paste with 4% NaCl was the most optimized for 3D printing, closely resembling the original template design with high accuracy.

#### 3.2.3. Post-Processing Properties

Table 7 provides insights into the cooking loss, transversal, and longitudinal shrinkage rates of the CS–MC gel with varying NaCl contents after heating. The incorporation of NaCl resulted in a significant reduction in cooking loss. Notably, the control group exhibited the highest cooking loss at 22.7%, while the sample with 4% NaCl exhibited the lowest, with a cooking loss of 18%. Typically, cooking loss is primarily attributed to moisture evaporation during heating [27]. The presence of NaCl forms a more complex network that prevents structural deformation and reduces moisture loss, leading to enhanced moisture retention (i.e., syneresis) in the gel [48]. This effect can be attributed to NaCl forming hydrogen bonds that limit moisture mobility, resulting in higher moisture retention compared to the control, as reported by Tan et al. [50].

Regarding shrinkage, all samples exhibited shrinkage rates within 10%, with a notable difference observed in transversal shrinkage. The control group exhibited higher shrinkage compared to the samples with NaCl additions. This increased shrinkage in the control group can be attributed to moisture loss as its structure was more prone to deformation, causing moisture to be pushed out due to weak network formation [51]. In summary, the addition of NaCl effectively reduces cooking loss and minimizes shape shrinkage by preventing moisture loss during cooking, offering superior protection against heating compared to the control.

#### 3.2.4. Mechanical Properties

Texture is a significant aspect of food that significantly influences consumer acceptance and mouthfeel [30]. Table 8 provides an overview of the textural properties of the 3D-printed samples with varying NaCl concentrations.

The hardness of the sample containing 4% NaCl was notably higher at 220.57 N compared to the control sample, which exhibited a hardness of 124.55 N. This increase in hardness can be attributed to the heat generated during heating, which led to stronger bonding of salt ions in the solution to the polymer and the formation of a more rigid protein network structure [31]. Dexter et al. [52] reported that salt enhances the development of the protein network in wheat noodles, resulting in a strong and highly resistant wheat noodle. Although there is no gluten in RF, we believe that the protein present in RF, combined with salt, would contribute to the augmentation in the hardness of the samples. It has been reported that RF can contain a protein content as high as 7% [53]. Sangpring et al. [54] examined the effect of salt on the texture of rice noodles and reported that the presence of NaCl improved the formation of the protein network but led to a decrease in the packing of starch in the dehydrated rice noodles.

Adhesiveness, indicating the extent to which a sample adheres to a probe, showed no significant difference among the samples. Cohesiveness, a measure of the strength of the internal structure of food, tended to approach a value of 1 as the NaCl content increased. This suggests that NaCl acted as a cross-linker, promoting the formation of a robust internal network within the 3D-printed CS–MC–NaCl gel [55]. Springiness, which contributes to the maintenance of the final product’s shape, tended to show an increase with increasing NaCl concentration. This enhancement in springiness is likely due to the higher NaCl content facilitating more effective cross-linking between starch molecules in the paste, resulting in improved heat stability and increased elasticity [49]. Sangpring et al. [54] reported that the elongation of the cooked rice noodles showed a significant increase as the NaCl concentration increased. They attributed this observation to the elevated extensibility, possibly due to the well-established protein network in rice noodles with NaCl. Chewiness, representing the energy required for swallowing, showed a significant difference between the control and the samples with added NaCl. As expected, chewiness increased with higher NaCl concentrations. It is worth noting that there is a known correlation between chewiness and springiness [56].

The shear force obtained from cutting tests is analogous to the sensation experienced when initially cutting food with one’s front teeth. It serves as a common method for evaluating the strength of food materials [57]. As shown in Table 8, the shear force exhibited a notable increase corresponding to the quantity of NaCl added. This suggests that the paste structure with NaCl additions possesses a denser and more interconnected network structure [50].

## 4. Conclusions

The results obtained from this study are useful because a mixture of CS–MC with NaCl can serve various purposes, including acting as a thickening agent, facilitating gel formation, and inducing ice crystal formation in the food industry due to their unique properties. The addition of NaCl to a mixture of CS–MC can have several effects on the properties and behavior of the mixture, depending on the concentration of NaCl and the ratios of the CS–MC components. In the current study, the effects of NaCl on the physical properties of CS–MC mixtures in a food model system and CS–MC pastes/gels produced using a 3D food printer were comprehensively examined. Interestingly, divergent trends were noted: the introduction of NaCl resulted in a reduction of viscosity in the liquid-based CS–MC blends while concurrently fostering a more stable and robust structure in the 3D-printed and cooked CS–MC gels, characterized by their solid nature. We conclude that NaCl interfered with the ability of starch molecules in CS–MC blends to form a stable gel network, resulting in a decrease in viscosity. Conversely, NaCl could strengthen the protein network in CS–MC gels, leading to an increase in the stability or hardness of the cooked products. This indicates that more comprehensive studies should be conducted to understand the complex interaction of the effect of NaCl on starch-based foods containing hydrocolloids. Key variables to consider could include various types of starch or hydrocolloids and the type of food matrices.

## Figures and Tables

**Figure 1 foods-12-04390-f001:**
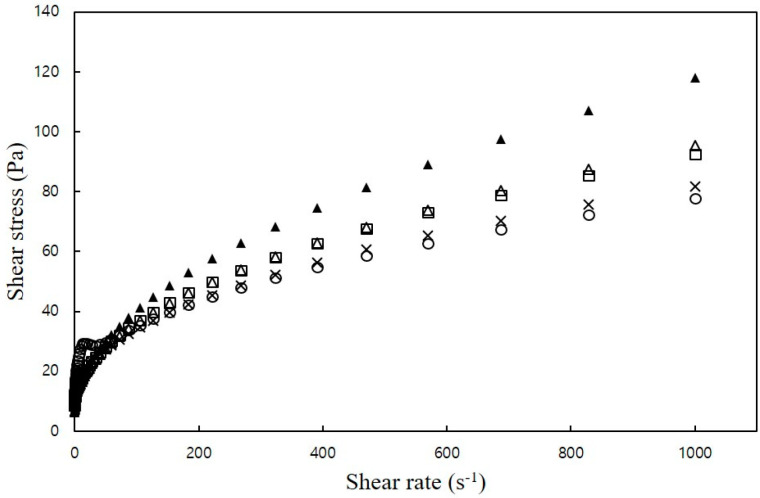
Plots of shear stress versus shear rate of CS–MC mixtures with different NaCl concentrations (○: 0% NaCl, ×: 1% NaCl, □: 2% NaCl, ∆: 3% NaCl, and ▲: 4% NaCl).

**Figure 2 foods-12-04390-f002:**
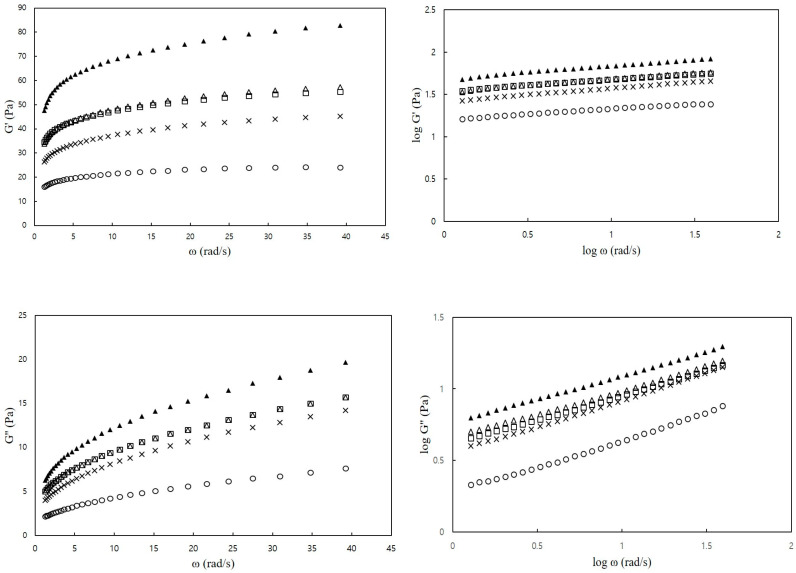
Plots of G′ and G″ versus ω (frequency) of CS–MC mixtures with different NaCl concentrations (○: 0% NaCl, ×: 1% NaCl, □: 2% NaCl, ∆: 3% NaCl, and ▲: 4% NaCl).

**Figure 3 foods-12-04390-f003:**
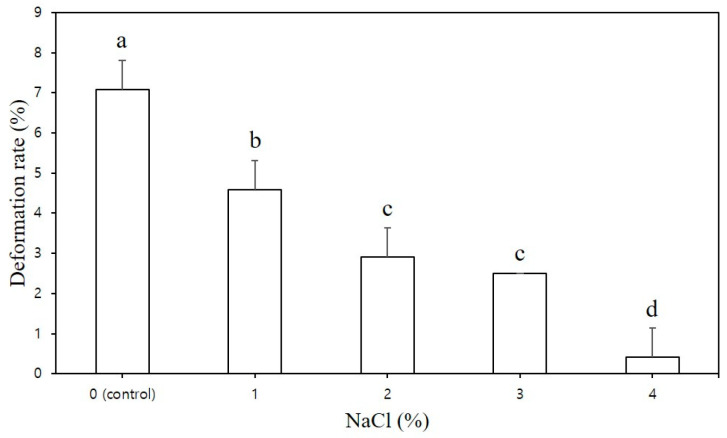
Deformation rate of the 3D-printed CS–MC paste with different NaCl concentrations. Different letters above the bar indicate significant differences among samples (*p* < 0.05).

**Figure 4 foods-12-04390-f004:**
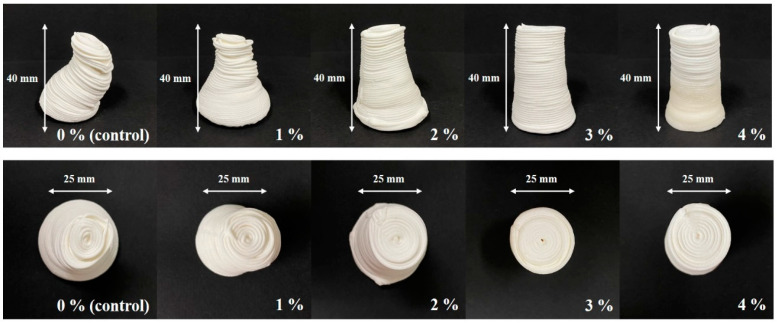
Dimensional stability images of the 3D-printed CS–MC paste with different concentrations of NaCl.

**Figure 5 foods-12-04390-f005:**
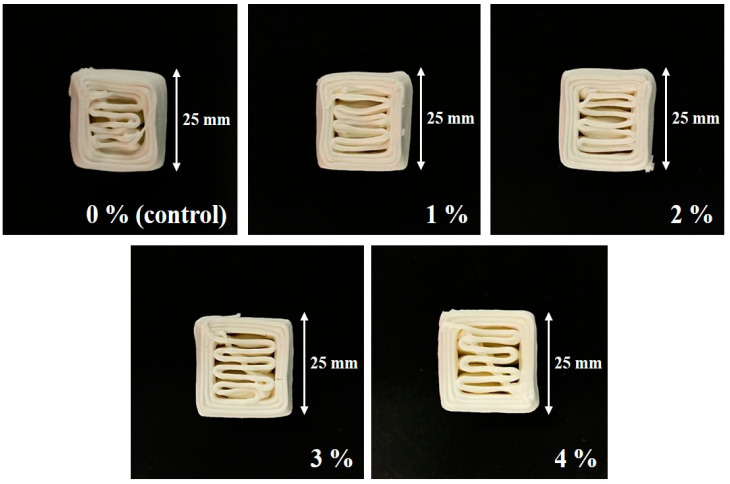
Images for the printing accuracy of the 3D-printed CS–MC paste with different concentrations of NaCl.

**Table 1 foods-12-04390-t001:** Composition of CS–MC pastes (*w*/*w*).

Sample	Corn Starch (g)	Methyl Cellulose (g)	NaCl (g)	Rice Flour (g)	Distilled Water (g)	Total Mass (g)
^1^ S0	17	0.5	0	40	42.5	100
S1	17	0.5	1	39	42.5	100
S2	17	0.5	2	38	42.5	100
S3	17	0.5	3	37	42.5	100
S4	17	0.5	4	36	42.5	100

^1^ CS–MC mixtures with different NaCl concentrations: S0: 0%, S1: 1%, S2: 2%, S3: 3%, and S4: 4% NaCl.

**Table 2 foods-12-04390-t002:** Swelling power of CS–MC mixtures with different NaCl concentrations.

NaCl (%)	Swelling Power (g/g)
0	^1^ 24.82 ± 0.40 ^a2^
1	12.75 ± 0.01 ^b^
2	8.90 ± 0.02 ^c^
3	6.81 ± 0.01 ^d^
4	5.53 ± 0.01 ^e^

^1^ Mean ± standard deviation. ^2^ Means with different letters in the column are significantly different (*p* < 0.05).

**Table 3 foods-12-04390-t003:** Pasting properties of CS–MC mixtures with different NaCl concentrations.

NaCl (%)	Pasting Temperature (°C)	Viscosity (cP)				
		Peak	Trough	Breakdown	Final	Setback
0	^1^ 88.8 ± 0.0 ^b2^	781.0 ± 4.2 ^a^	598.0 ± 59.4 ^a^	183.0 ± 55.2 ^a^	586.5 ± 55.5 ^a^	14.0 ± 1.8 ^a^
1	79.5 ± 0.6 ^c^	644.5 ± 7.7 ^b^	477.0 ± 1.4 ^b^	167.5 ± 9.1 ^ab^	490.5 ± 0.7 ^a^	13.5 ± 2.1 ^a^
2	90.8 ± 0.4 ^b^	592.5 ± 6.3 ^c^	488.5 ± 9.1 ^b^	104.0 ± 2.8 ^abc^	506.0 ± 5.6 ^a^	17.5 ± 0.7 ^a^
3	91.7 ± 1.6 ^ab^	549.0 ± 0.0 ^d^	478.5 ± 0.7 ^b^	70.5 ± 0.7 ^bc^	500.0 ± 1.4 ^a^	21.5 ± 2.1 ^a^
4	94.8 ± 0.4 ^a^	533.5 ± 10.6 ^d^	479.5 ± 10.6 ^b^	54.0 ± 0.0 ^c^	501.0 ± 12.7 ^a^	21.5 ± 2.1 ^a^

^1^ Mean ± standard deviation. ^2^ Means with different letters in the same column for each parameter are significantly different (*p* < 0.05).

**Table 4 foods-12-04390-t004:** Steady shear rheological properties of CS–MC mixtures with different NaCl concentrations.

NaCl (%)	Power Law Model					Casson Model
	Flow Behaviour Index, n	Consistency Index, K (Pa·s^n^)	Apparent Viscosity, η_a,500_ (Pa·s)	R^2^	Casson Yield Stress, σ_oc_ (Pa)	R^2^
0	^1^ 0.20 ± 0.00 ^c2^	16.04 ± 0.74 ^a^	0.11 ± 0.01 ^b^	0.97	16.87 ± 0.65 ^a^	0.92
1	0.22 ± 0.00 ^bc^	13.49 ± 0.72 ^ab^	0.11 ± 0.01 ^b^	0.96	13.56 ± 0.42 ^b^	0.97
2	0.25 ± 0.01 ^bc^	12.35 ± 1.14 ^b^	0.11 ± 0.00 ^b^	0.95	12.16 ± 1.07 ^b^	0.97
3	0.27 ± 0.05 ^b^	10.86 ± 2.23 ^bc^	0.12 ± 0.01 ^b^	0.95	10.69 ± 2.22 ^bc^	0.98
4	0.34 ± 0.01 ^a^	8.84 ± 0.05 ^c^	0.14 ± 0.01 ^a^	0.98	8.83 ± 0.28 ^c^	0.97

^1^ Mean ± standard deviation. ^2^ Means with different letters in the same column for each parameter are significantly different (*p* < 0.05).

**Table 5 foods-12-04390-t005:** The storage modulus (G′) and loss modulus (G″), complex viscosity (η*), and tan δ at 6.28 rad/s of CS–MC mixtures with different NaCl concentrations.

NaCl (%)	Storage Modulus G′ (Pa)	Loss Modulus G″ (Pa)	Complex Viscosity η* (Pa·s)	Tan δ
0	^1^ 19.95 ± 0.81 ^c2^	3.66 ± 0.20 ^c^	3.22 ± 0.12 ^c^	0.18 ± 0.02 ^ab^
1	34.58 ± 6.89 ^b^	7.02 ± 1.26 ^b^	5.60 ± 1.11 ^b^	0.20 ± 0.00 ^a^
2	44.09 ± 2.57 ^b^	7.66 ± 0.49 ^b^	7.10 ± 0.39 ^b^	0.17 ± 0.02 ^ab^
3	44.49 ± 1.59 ^b^	8.31 ± 0.57 ^ab^	7.18 ± 0.26 ^b^	0.19 ± 0.01 ^ab^
4	63.50 ± 7.54 ^a^	10.59 ± 1.51 ^a^	10.22 ± 1.22 ^a^	0.17 ± 0.00 ^b^

^1^ Mean ± standard deviation. ^2^ Means with different letters in the same column for each parameter are significantly different (*p* < 0.05).

**Table 6 foods-12-04390-t006:** Printing accuracy of the 3D-printed CS–MC paste with different NaCl concentrations.

NaCl (%)	Length (%)	Width (%)	Height (%)
0	^1^ 100.0 ± 0.0 ^a2^	100.0 ± 1.2 ^a^	97.8 ± 2.0 ^a^
1	100.0 ± 1.2 ^b^	100.0 ± 1.2 ^b^	98.9 ± 2.0 ^a^
2	100.0 ± 1.2 ^b^	100.0 ± 1.2 ^b^	98.9 ± 2.0 ^a^
3	100.0 ± 1.2 ^b^	100.0 ± 1.2 ^b^	98.9 ± 2.0 ^a^
4	100.0 ± 0.0 ^b^	100.0 ± 0.0 ^b^	100.0 ± 0.0 ^a^

^1^ Mean ± standard deviation. ^2^ Means with different letters in the same column are significantly different (*p* < 0.05).

**Table 7 foods-12-04390-t007:** Shrinkage and cooking loss of the 3D-printed CS–MC gel with different concentrations of NaCl.

NaCl (%)	Cooking Loss (%)	Transversal Shrinkage Ratio (%)	Longitudinal Shrinkage Ratio (%)
0	^1^ 22.74 ± 0.84 ^a2^	9.59 ± 0.76 ^a^	6.63 ± 0.36 ^a^
1	20.50 ± 0.41 ^b^	4.98 ± 0.43 ^b^	5.14 ± 0.52 ^a^
2	19.09 ± 0.71 ^c^	4.30 ± 0.47 ^b^	5.16 ± 0.28 ^a^
3	18.98 ± 0.33 ^c^	4.02 ± 0.94 ^b^	5.22 ± 0.30 ^a^
4	17.98 ± 0.33 ^d^	3.69 ± 0.74 ^b^	5.30 ± 0.08 ^a^

^1^ Mean ± standard deviation. ^2^ Means with different letters in the same column are significantly different (*p* < 0.05).

**Table 8 foods-12-04390-t008:** Textural properties of the 3D-printed CS–MC gel with different concentrations of NaCl.

NaCl (%)	Hardness (N)	Adhesiveness (N·sec)	Cohesiveness (-)	Springiness (-)	Chewiness (-)	Cutting Force (N)
0	^1^ 124.55 ± 6.42 ^b2^	−0.7 ± 0.98 ^a^	0.64 ± 0.01 ^b^	0.8. ± 0.05 ^a^	175.7 ± 58.4 ^b^	64.25 ± 4.96 ^c^
1	142.63 ± 22.07 ^b^	0.3 ± 0.92 ^a^	0.67 ± 0.01 ^b^	0.81 ± 0.01 ^a^	280.2 ± 61.5 ^b^	78.27 ± 11.82 ^c^
2	174.33 ± 5.36 ^b^	−0.1 ± 1.23 ^a^	0.68 ± 0.01 ^b^	0.84 ± 0.01 ^a^	1114.8 ± 148.3 ^a^	102.15 ± 5.56 ^bc^
3	184.07 ± 11.67 ^b^	−0.5 ± 1.07 ^a^	0.73 ± 0.01 ^a^	0.84 ± 0.01 ^a^	1150.8 ± 261.7 ^a^	111.67 ± 7.93 ^b^
4	220.57 ± 2.28 ^a^	−0.5 ± 0.91 ^a^	0.71 ± 0.01 ^a^	0.85 ± 0.03 ^a^	1177.2 ± 111.9 ^a^	133.73 ± 8.1 ^a^

^1^ Mean ± standard deviation. ^2^ Means with different letters in the same column are significantly different (*p* < 0.05).

## Data Availability

Data are contained within the article.

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
