# Peer review of "Effects of NaCl on the Physical Properties of Cornstarch–Methyl Cellulose Blend and on Its Gel Prepared with Rice Flour in a Model System"

_foods, 2023, doi:10.3390/foods12244390_

Round 1
Reviewer 1 Report (Previous Reviewer 3)
Comments and Suggestions for Authors
The revised version is acceptable.
Reviewer 2 Report (Previous Reviewer 2)
Comments and Suggestions for Authors
I sincerely thank the authors for the time spent in revising their manuscript and for replying in a satisfactory way to my questions and doubts.
I have no further comments.
Comments on the Quality of English Language
The english language can be improved in some parts.
This manuscript is a resubmission of an earlier submission. The following is a list of the peer review reports and author responses from that submission.
Round 1
Reviewer 1 Report
Comments and Suggestions for Authors
Comments on FOODS_2670655 entitled ‘Effects of NaCl on the physical properties of corn starch-methyl cellulose blends and pastes made using 3D food printer'.
The study has some weaknesses that should be addressed before being suitable for a possible publication
First main issue regards the aim of the study and experimental design. It is not clear why the authors choose the 3D food printing technology and how they contribute to the ‘practical application’. All the study could also be conducted with a traditional technology or molding. There is no study on printing variables.
Also, the authors analysed some physical properties on MC-CS mixture with added salt while they printed a different paste in which the main ingredient was rice. This does not allow a good understanding of the data presented, due to the fact that there is no relationship among analysis before and after printing. Also, salt could exert an effect on rice and this aspect was not considered in the discussion of data.
Other suggestions:
The introduction section should be improved on what is already known about this topic.
I suggest adding a correlation analysis to add useful information.
Minor suggestions:
Concentration of MC-CS mixture is 5% as reported in lines 88 or 121, or 0.5% as in line 106?
Generally, in 3D printing the nozzle or layer height is 80% or equal to the nozzle diameter. This allows a good deposition and adhesion of deposited filaments, widely recognized. Why in this study the nozzle height is higher (1.8 mm) in respect to the nozzle diameter (1.1 mm)?
Why were different shapes printed? And used for different analyses?
Which of these shapes is then cooked?
Better to use RVU than cP as units for pasting properties
Information about repetition of all the analysis is missed
How did the authors select the printing parameters? On which food formula? It can be hypothesized that the pastes, having different physical properties, behave differently during printing.
Hardness values should be reported in N
Author Response
Comments on FOODS_2670655 entitled ‘Effects of NaCl on the physical properties of corn starch-methyl cellulose blends and pastes made using 3D food printer'.
The study has some weaknesses that should be addressed before being suitable for a possible publication. First main issue regards the aim of the study and experimental design. It is not clear why the authors choose the 3D food printing technology and how they contribute to the ‘practical application’. All the study could also be conducted with a traditional technology or molding. There is no study on printing variables.
First of all, the authors deeply appreciate all the reviewers’ comments to improve the current manuscript. We did our best to respond to reviewers’ comments point-by-point.
To better clarify the objective of the current study, in revised version, we divided it into two studies (i.e., experiments). The first study focused on investigating the effect of NaCl on the physical properties of CS-MC blends more likely in liquid form. The second study focused on assessing the physical properties of the solid-based gel prepared with rice flour made using the same CS-MC blends ratio used in the first experiment.
Since the objective of using a 3D printer in the second study of this manuscript is to produce a gel in a model system and assess the physical properties of the gels, we revised the title and objective of this current manuscript. The title was revised as follows. Effects of NaCl on the physical properties of corn starch-methyl cellulose blend and on its gel prepared with rice flour in model system.
In addition, we strongly believe that the first experiment represents a fundamental study on the physical properties of CS-MC blends. In contrast, the second experiment is more aligned with the practical application based on these blends. This is evident as solid gels prepared with rice flour and water, incorporating the same CS-MC blends were produced using 3D printing. The term 3D printed 'paste' was changed to 'gel' after the paste was cooked to better denote the solid-based samples. This statement was added to the manuscript.
Please see lines 225-228.
The authors also believe that the dimensional stability of the CS-MC gel cannot be efficiently measured with traditional technology or molding but can be more accurately assessed through 3D printing. This type of stacked specimen with thin layers is considered optimized for evaluating the dimensional stability of the CS-MC gel. Please refer to the current manuscript’s Fig 5. Dimensional stability images of the 3D-printed CS-MC paste with different concentration of NaCl.
Also, the authors analyzed some physical properties on MC-CS mixture with added salt while they printed a different paste in which the main ingredient was rice. This does not allow a good understanding of the data presented, due to the fact that there is no relationship among analysis before and after printing. Also, salt could exert an effect on rice and this aspect was not considered in the discussion of data.
As described above, we wanted to produce a simple solid-based gels like a model system for the practical application using the MC-CS blends. Among many candidates to help produce a solid-based sample, we thought rice flour could be a good candidate as it is a versatile ingredient commonly used in the processing of various food products.
Please see lines……..
Discussion regarding the interaction of salt and rice flour have been addressed in a revised version of manuscript.
In some literatures, rice flour is suitable for 3D printing in the creation of food products with specific shapes and structures.
< Pulatsu, E., Su, J. W., Lin, J., & Lin, M. (2020). Factors affecting 3D printing and post-processing capacity of cookie dough. Innovative Food Science & Emerging Technologies, 61, 102316.>
< Anukiruthika, T., Moses, J. A., & Anandharamakrishnan, C. (2020). 3D printing of egg yolk and white with rice flour blends. Journal of Food Engineering, 265, 109691.)
< Bhat, Z. F., Morton, J. D., Kumar, S., Bhat, H. F., Aadil, R. M., & Bekhit, A. E. D. A. (2021). 3D printing: Development of animal products and special foods. Trends in Food Science & Technology, 118, 87-105.)
Other suggestions:
The introduction section should be improved on what is already known about this topic.
The introduction section has been extensively revised according to reviewer’s comments.
Please see the Introduction section.
I suggest adding a correlation analysis to add useful information.
Other reviewers have pointed out that there are too many tables and figures in the current manuscript. Unless significant improvements are not achieved or it is not mandatory to add a correlation analysis, the authors would prefer not to conduct the correlation analyses. Please understand this.
Minor suggestions:
Concentration of MC-CS mixture is 5% as reported in lines 88 or 121, or 0.5% as in line 106?
There was a typo and that it is corrected.
Please see line 165.
Generally, in 3D printing the nozzle or layer height is 80% or equal to the nozzle diameter. This allows a good deposition and adhesion of deposited filaments, widely recognized. Why in this study the nozzle height is higher (1.8 mm) in respect to the nozzle diameter (1.1 mm)?
Please refer to the following response. We conducted numerous preliminary tests to achieve stable starch paste/gels, leading us to the conclusion of using a specific ratio between the nozzle height (1.8 mm) and the nozzle diameter (1.1 mm). This decision was based on the close proximity between the starch paste printed by the 3D printer and its nozzle. The proximity was such that the nozzle kept touching the printed paste, causing it to collapse from its original shapes.
Relevant statements were included in the manuscript.
Please see lines 264-273.
Why were different shapes printed? And used for different analyses?
Two different shapes of starch paste printed using 3D printer were employed in this study. The cuboid-shaped paste was utilized for assessing printing accuracy and underwent post-cooking, including all textural analyses. Conversely, the cylindrical-shaped paste was exclusively employed for only dimensional stability measurements. To enhance clarity, we have incorporated relevant descriptions.
Please see lines 250-254.
In addition, for a smoother flow in the manuscript, we have rearranged the sections, placing dimensional stability before printing accuracy in both the Materials and Methods, as well as the Results and Discussion sections. This adjustment allows for a more logical discussion of pastes/gels with a cuboid shape after post-cooking, incorporating all texture analyses.
Which of these shapes is then cooked?
Please refer to the right above question.
Better to use RVU than cP as units for pasting properties
We are sorry for the inconvenience, but we consistently encounter software errors when attempting to convert units from cP to RVU for the pasting properties. We are unsure of this reason. Kindly permit us to retain the original unit for this purpose.
Information about repetition of all the analysis is missed.
It was corrected.
How did the authors select the printing parameters? On which food formula? It can be hypothesized that the pastes, having different physical properties, behave differently during printing.
The 3D printing parameters we used in this study was mostly based on the following paper.
< Ko, H. J., Wen, Y., Choi, J. H., Park, B. R., Kim, H. W., & Park, H. J. (2021). Meat analog production through artificial muscle fiber insertion using coaxial nozzle-assisted three-dimensional food printing. Food Hydrocolloids, 120, 106898>
We thought that our starch paste with 3D printing could be as similar as the ISP (isolated soy protein) used for meat analog in the above reference. Accordingly, we did many preliminary experiments to find optimal 3D printing parameters.
Hardness values should be reported in N
It was converted. Please see table 8.
Reviewer 2 Report
Comments and Suggestions for Authors
General comment:
This work deals with the characterization of 3D-printed pastes of corn starch-methyl cellulose mixtures when NaCl concentration is varying. The work is aimed at the identifying suitable mixtures and formulations that are printable.
Specific comments throughout the paper:
1.
Please provide quantitative details about the shear and heat parameters affecting processing conditions of native starch.
Line 57-60: missing details about the rheological properties.
Lines 64-77: the introduction is poorly focused to the aim and scope of the work. Furthermore, the authors are not dealing and analyzing properly the most recent works. Intro must be revised.
2.
Line91-93: provide more details about this relevant point.
Lines 114-117: missing discussion, justification and demonstration for the usability of this method
Please format Eq. (1) and (2) as per template instructions. Also use the same symbols between eq. And text.
Any statistical tests on the rheological measurements?
Lines 178: please provide clearly which are the modifications
Dimensional stability is something like
1 - 1/printing accuracy
How did the authors select these figures of merit?
Several characterization have been carried out. This is remarkable.
3.
Results presentation is somehow smooth and a good commenting has been provided, with some discussion and comparison with other references.
An inset in Fig. 1 can be provided to zoom and allow the visualization of the data in the [0,100] 1/s shear rate region.
The rheological results are confirmed by other literature studies.
A comparison of the rheological parameters with the literature data is missing.
Tab. 6 is more useful if the standard deviation is considered. The authors should analyze it rather than considering the mean value.
Fig. 4 has a small font, please increase it.
The post-processing and mechanical characterization data are valid.
4.
Conclusion section is fine.
Edit & formatting: The manuscript must be proofread and the formatting style should be compliant to the template (e.g., the text is not justified and properly aligned). Please check and verify.
Comments on the Quality of English Language
The english language demands minor corrections
Author Response
This work deals with the characterization of 3D-printed pastes of corn starch-methyl cellulose mixtures when NaCl concentration is varying. The work is aimed at the identifying suitable mixtures and formulations that are printable.
Specific comments throughout the paper:
- Please provide quantitative details about the shear and heat parameters affecting processing conditions of native starch.
The following paragraph was added to the manuscript. Namely, native starch can be sensitive to shear forces. Excessive mechanical processing or shear can lead to breakdown and a reduction in viscosity. Additionally, it may not withstand high temperatures without undergoing irreversible changes such as gelatinization or retrogradation.
Please see lines 47-51.
Line 57-60: missing details about the rheological properties.
Both starch and hydrocolloids influence the rheological properties of food, and their impact is significantly influenced by factors such as type and concentration, the ratio of amylose to amylopectin, pH, and the presence of other components like sugars or salts [18].
Relevant information was added to the manuscript.
Please see lines 66-67.
Lines 64-77: the introduction is poorly focused to the aim and scope of the work. Furthermore, the authors are not dealing and analyzing properly the most recent works. Intro must be revised.
As the reviewer pointed out, the introduction section has extensively revised. Please see the introduction section.
- Line91-93: provide more details about this relevant point.
The relevant description was added to the manuscript.
Please see lines 147-150.
Lines 114-117: missing discussion, justification and demonstration for the usability of this method
The swelling power of the paste was mostly determined with reference to the following source. We believe that measuring the swelling power of the 3D printed paste is crucial as it indicates the starch granule's ability to absorb water and the extent of amylose leaching out during swelling.
The relevant description was added to the manuscript.
Please see lines 162-165.
<Wang, W.; Zhou, H.; Yang, H.; Zhao, S.; Liu, Y.; Liu, R. Effects of salts on the gelatinization and retrogra-dation properties of maize starch and waxy maize starch. Food Chemistry 2017, 214, 319-327. https://doi.org/10.1016/j.foodchem.2016.07.040.>
Please format Eq. (1) and (2) as per template instructions. Also use the same symbols between eq. And text.
It was corrected. Please see Eq. 1 and 2.
Any statistical tests on the rheological measurements?
The data for pasting properties obtained from the RVA were statistically analyzed for mean separation. The data (flow behavior, consistency index, and apparent viscosity) for Fig. 1 (plot of shear stress versus shear rate of CS-MC mixtures with different NaCl concentrations) were statistically analyzed for mean separation, as shown in Table 4. The data (storage modulus, loss modulus, and complex viscosity) for Fig. 2 (Plots of G’ and G’’ versus ω (frequency) of CS-MC mixtures with different NaCl concentrations) were statistically analyzed for mean separation, as shown in Table 5.
Lines 178: please provide clearly which are the modifications.
Since the printed paste differed from that in the reference, most modifications were made to the printing parameters. The relevant statements have been added to the manuscript as follows.
The 3D printing of starch paste with various concentrations of NaCl was conducted using a 3D food printer (YOLI-LAB, YOLILO Co., Ltd., Korea), following the methods described by Ko et al. [24] with slight modifications, particularly in the printing parameters such as nozzle dimension, infill level, and pattern.
Please see lines 253-254.
Dimensional stability is something like 1 - 1/printing accuracy
Yes, the dimensional stability and printing accuracy are somewhat similarly calculated. Please refer to the following responses for the justification of the use of these two measurements.
Two different shapes of starch paste were employed in this study. The cuboid-shaped paste was utilized for assessing printing accuracy and underwent post-cooking, including all textural analyses. Conversely, the cylindrical-shaped paste was exclusively employed for dimensional stability measurements. To enhance clarity, we have incorporated relevant descriptions.
Please see lines 250-254.
In addition, for a smoother flow in the manuscript, we have rearranged the sections, placing dimensional stability before printing accuracy in both the Materials and Methods, as well as the Results and Discussion sections. This adjustment allows for a more logical discussion of pastes with a cuboid shape after post-cooking, incorporating all texture analyses.
How did the authors select these figures of merit?
Can you please specify the figures that you mentioned?
Several characterizations have been carried out. This is remarkable.
We appreciate your comments.
- Results presentation is somehow smooth and a good commenting has been provided, with some discussion and comparison with other references.
We appreciate your comments.
An inset in Fig. 1 can be provided to zoom and allow the visualization of the data in the [0,100] 1/s shear rate region.
We are currently encountering technical errors with the Microsoft Word document. Therefore, any issues such as configuring figures, text alignment, or line numbers in the authors' Microsoft Word document will be addressed and corrected by the Journal editorial office upon manuscript acceptance. We appreciate your understanding.
The rheological results are confirmed by other literature studies.
We appreciate your comments.
A comparison of the rheological parameters with the literature data is missing.
Results of the rheological parameters in the current study were compared and discussed with other references throughout the manuscript, mostly with the following manuscript because it examined the effect of NaCl on gelatinization and rheological properties of sweet potato starch–xanthan gum mixture.
<Gil, B., & Yoo, B. (2015). Effect of salt addition on gelatinization and rheological properties of sweet potato starch–xanthan gum mixture. Starch‐Stärke, 67(1-2), 117-123.>
Please see the results and discussion in the revised version of manuscript.
Tab. 6 is more useful if the standard deviation is considered. The authors should analyze it rather than considering the mean value.
Thank you for your keen insights. It was also observed that the standard deviation for the length, width, and height values of the final printed product tended to decrease as the concentration of NaCl increased. This suggests that the addition of NaCl contributed to increased stability in the printed products. This statement was added to the manuscript.
Please see lines 7-11 on page 17.
Fig. 4 has a small font, please increase it.
We are currently encountering technical errors with the Microsoft Word document. Therefore, any issues such as configuring figures, text alignment, or line numbers in the authors' Microsoft Word document will be addressed and corrected by the Journal editorial office upon manuscript acceptance. We appreciate your understanding.
The post-processing and mechanical characterization data are valid.
We appreciate your comments.
- Conclusion section is fine.
We appreciate your comments.
Edit & formatting: The manuscript must be proofread and the formatting style should be compliant to the template (e.g., the text is not justified and properly aligned). Please check and verify.
We are currently encountering technical errors with the Microsoft Word document. Therefore, any issues such as configuring figures, text alignment, or line numbers in the authors' Microsoft Word document will be addressed and corrected by the Journal editorial office upon manuscript acceptance. We appreciate your understanding.
Reviewer 3 Report
Comments and Suggestions for Authors
The authors added NaCl in corn starch-methyl cellulose blends and tested the pasting, rheology of the blends. And CS-MC pastes were prepared for 3D printing. Even though a lot of data was shown in the draft, it didn’t reflect the expertise and carefulness of the authors.
1. The biggest problem is the connection between rheology test and 3D printing. I can’t see the connection between them because the samples used in these parts were different. The rheology should be conducted on the CS-MC pastes to explain the 3D printing results. Additional experiments should be done.
2. Title: the title should be revised to make it clear.
3. Abstract: the authors stated that NaCl could enhance protein network within CS-MC pastes. But I searched the manuscript, they just mentioned protein network in abstract and conclusion. How can they get the conclusion? Where is the protein network from? What’s the interaction between NaCl and rice flour in the pastes?
4. Introduction: I can’t see the novelty of the study in the introduction. There are numerous manuscripts about 3D printing of hydrocolloids, what’s new of your study? What’s the application of your study?
5. The data in the manuscript is not convincing due to the abnormal results of the hardness. I can’t believe the hardness of a 3D printing pasts is up to 30,000 g. The authors should reconsider the result and repeat it.
6. I can’t see too much about the poor discussion of the manuscript, it seems like the sections can’t be connected.
Comments on the Quality of English Language
The English should be improved.
Author Response
The authors added NaCl in corn starch-methyl cellulose blends and tested the pasting, rheology of the blends. And CS-MC pastes were prepared for 3D printing. Even though a lot of data was shown in the draft, it didn’t reflect the expertise and carefulness of the authors.
- The biggest problem is the connection between rheology test and 3D printing. I can’t see the connection between them because the samples used in these parts were different. The rheology should be conducted on the CS-MC pastes to explain the 3D printing results. Additional experiments should be done.
First of all, the authors deeply appreciate all the reviewers’ comments to improve the current manuscript. We did our best to respond to reviewers’ comments point-by-point.
To better clarify the objective of the current study, in revised version, we divided it into two studies (i.e., experiments). The first study focused on investigating the effect of NaCl on the physical properties of CS-MC blends more likely in liquid form. The second study focused on assessing the physical properties of the solid-based gel prepared with rice flour made using the same CS-MC blends ratio used in the first experiment.
Since the objective of using a 3D printer in the second study of this manuscript is to produce a gel in a model system and assess the physical properties of the gels, we revised the title and objective of this current manuscript. The title was revised as follows. Effects of NaCl on the physical properties of corn starch-methyl cellulose blend and on its gel prepared with rice flour in model system.
In addition, we strongly believe that the first experiment represents a fundamental study on the physical properties of CS-MC blends. In contrast, the second experiment is more aligned with the practical application based on these blends. This is evident as solid gels prepared with rice flour and water, incorporating the same CS-MC blends were produced using 3D printing. The term 3D printed 'paste' was changed to 'gel' after the paste was cooked to better denote the solid-based samples. This statement was added to the manuscript.
Please see lines………………
The authors also believe that the dimensional stability of the CS-MC gel cannot be efficiently measured with traditional technology or molding but can be more accurately assessed through 3D printing. This type of stacked specimen with thin layers is considered optimized for evaluating the dimensional stability of the CS-MC gel. Please refer to the current manuscript’s Fig 5. Dimensional stability images of the 3D-printed CS-MC paste with different concentration of NaCl.
The authors denote these two terms: paste and gel. The term 'paste' refers to the samples printed using a 3D printer and not cooked, while the term 'gel' refers to the samples after the printed pastes were cooked.
Please see lines 225-228.
- Title: the title should be revised to make it clear.
The title was revised as follows. Effects of NaCl on the physical properties of corn starch-methyl cellulose blend and on its gel prepared with rice flour in model system.
- Abstract: the authors stated that NaCl could enhance protein network within CS-MC pastes. But I searched the manuscript, they just mentioned protein network in abstract and conclusion. How can they get the conclusion? Where is the protein network from? What’s the interaction between NaCl and rice flour in the pastes?
Please the revised results and discussion section, focusing on Table 8.
- Introduction: I can’t see the novelty of the study in the introduction. There are numerous manuscripts about 3D printing of hydrocolloids, what’s new of your study? What’s the application of your study?
The introduction section has been extensively revised according to reviewer’s comments.
Please see the Introduction section of the revised manuscript.
- The data in the manuscript is not convincing due to the abnormal results of the hardness. I can’t believe the hardness of a 3D printing pasts is up to 30,000 g. The authors should reconsider the result and repeat it.
We repeated experiments, obtaining new values that were converted into Newtons.
Just for the clarity, two different shapes of starch paste were employed in this study. The cuboid-shaped paste was utilized for assessing printing accuracy and underwent post-cooking, including all textural analyses. Conversely, the cylindrical-shaped paste was exclusively employed for dimensional stability measurements. To enhance clarity, we have incorporated relevant descriptions.
Please see lines 250-254.
In addition, for a smoother flow in the manuscript, we have rearranged the sections, placing dimensional stability before printing accuracy in both the Materials and Methods, as well as the Results and Discussion sections. This adjustment allows for a more logical discussion of pastes with a cuboid shape after post-cooking, incorporating all texture analyses.
- I can’t see too much about the poor discussion of the manuscript, it seems like the sections can’t be connected.
Discussion sections were extensively revised and improved.
Please see the results and discussion section of the revised manuscript.
Round 2
Reviewer 3 Report
Comments and Suggestions for Authors
I insist my decision based on the revised version mainly because I can't see the expertise in the response and revised version.
(1) In the response, the author mentioned "This statement was added to the manuscript.
Please see lines………………
The authors also believe that the dimensional stability of the CS-MC gel cannot be efficiently measured with traditional technology or molding but can be more accurately assessed through 3D printing. "
But where is the lines?
(2) The didn't response to this point : Abstract: the authors stated that NaCl could enhance protein network within CS-MC pastes. But I searched the manuscript, they just mentioned protein network in abstract and conclusion. How can they get the conclusion? Where is the protein network from? What’s the interaction between NaCl and rice flour in the pastes? And this point: Please the revised results and discussion section, focusing on Table 8.
(3) The revised part were not addressed in red. I can only see a small part of it.
Overall, the revised version didn’t reflect the expertise and carefulness of the authors.
Author's response:
I insist my decision based on the revised version mainly because I can't see the expertise in the response and revised version.
(1) In the response, the author mentioned "This statement was added to the manuscript.
Please see lines………………
I have deliberately deleted this sentence because my responses are integrated throughout the manuscript.
The authors also believe that the dimensional stability of the CS-MC gel cannot be efficiently measured with traditional technology or molding but can be more accurately assessed through 3D printing. "
But where is the lines?
It was added to the manuscript. Please see lines 214-217.
(2) The didn't response to this point : Abstract: the authors stated that NaCl could enhance protein network within CS-MC pastes. But I searched the manuscript, they just mentioned protein network in abstract and conclusion. How can they get the conclusion? Where is the protein network from? What’s the interaction between NaCl and rice flour in the pastes? And this point: Please the revised results and discussion section, focusing on Table 8.
I have extensively revised the first manuscript using 'track changes' function in Microsoft Word.
(3) The revised part were not addressed in red. I can only see a small part of it.
I have extensively revised the first manuscript using 'track changes' function in Microsoft Word.
Overall, the revised version didn’t reflect the expertise and carefulness of the authors.